# Acute and Long-Term Effects of an Internet-Based, Self-Help Comprehensive Behavioral Intervention for Children and Teens with Tic Disorders with Comorbid Attention Deficit Hyperactivity Disorder, or Obsessive Compulsive Disorder: A Reanalysis of Data from a Randomized Controlled Trial

**DOI:** 10.3390/jcm11010045

**Published:** 2021-12-23

**Authors:** Lilach Rachamim, Hila Mualem-Taylor, Osnat Rachamim, Michael Rotstein, Sharon Zimmerman-Brenner

**Affiliations:** 1School of Psychology, Reichman University IDC Herzliya, Herzliya 4610101, Israel; hilm140@gmail.com (H.M.-T.); zbsharon@idc.ac.il (S.Z.-B.); 2Donald J. Cohen & Irving B. Harris Resilience Center, Association for Children at Risk, Tel Aviv 6719958, Israel; 3Pediatric Movement Disorders Clinic, Pediatric Neurology Unit, Dana-Dwek Children’s Hospital, Tel Aviv Sourasky Medical Center, Tel Aviv 6423906, Israel; osirach27@gmail.com (O.R.); michaelr@tlvmc.gov.il (M.R.)

**Keywords:** Tourette syndrome, tic disorders, attention deficit hyperactivity disorder, impulse control, obsessive compulsive disorder, obsessive compulsive behavior

## Abstract

Attention deficit hyperactivity disorder (ADHD), obsessive compulsive disorder (OCD) and tic disorders (TD) commonly co-occur. In addition, specific inattention difficulties and poor impulse control are related to TD in the absence of comorbid ADHD. In this study we reanalyzed data from a recently completed study comparing internet-delivered, self-help comprehensive behavioral intervention for tics (ICBIT) with a waiting-list control group. The current study describes the effects of an (ICBIT) in children and adolescents with TD with and without comorbid diagnoses of ADHD or OCD at post intervention and over three- and six-month follow-up periods. Thirty-eight 7 to 18-year-olds completed the ICBIT. Of these, 16 were diagnosed with comorbid ADHD and 11 were diagnosed with OCD. A significant improvement in tic measures was found in all groups. Both the TD + ADHD and the TD − ADHD groups were similar in the magnitude of tic reduction from baseline to post-treatment, and at the three and six-month follow-up assessments. However, the TD + OCD group benefitted less from intervention than the TD—OCD group. There were meaningful reductions in parental reports of inattention, as well as hyperactive and impulsive symptoms at post intervention and over the 6-month follow-up period. Thus, ICBIT can be effectively delivered in the presence of comorbid ADHD or OCD symptomatology and may reduce symptoms of inattention and impulsivity. Larger studies of ICBIT in children and teens with TD and comorbid ADHD and OCD are needed to optimize responses to ICBIT.

## 1. Introduction

Tourette syndrome (TS) and chronic tic disorder (CTD) (collectively referred to as tic disorders TD) are neurodevelopmental disorders characterized by sudden, rapid, recurrent motor movements or vocalizations that persist for more than a year [1]. TS is defined by the presence of multiple motor and phonic tics. Children who exhibit motor or phonic tics, but not both, for at least a year are diagnosed as having CTD [1]. Although tic symptoms remit for some children and teens, a considerable portion of adolescents have tics that persist into adulthood [2]. The population prevalence of CTD in children and teens is 1.5% to 3% and TS is estimated to occur in 0.3% to 0.9% of this population as a whole. Males are more commonly affected than females at a ratio of 3:1 to 4:1 [1,3]. TD can lead to behavioral and psychosocial impairments, loneliness, as well as shame and embarrassment [3,4].

People with TD, including children and adolescents, are able to suppress their tics for variable periods of time but frequently experience premonitory urges, which are defined as unpleasant and distressing sensory phenomena or perceived need to move that often increase before tics and during tic suppression. These premonitory urges are often momentarily relieved by the expression of the tic. The core therapeutic techniques of habit reversal training (HRT) and its expansion comprehensive behavioral intervention for tics (CBIT) for people with TD are forms of interoceptive awareness training and tic suppression that harness a competitive response based on this suppressibility. These interventions eventually lead to decreased tic frequency and severity [5,6].

Of all the comorbid disorders associated with TD, attention-deficit/hyperactivity disorder (ADHD) and obsessive compulsive disorder (OCD) and subclinical obsessive compulsive behavior (OCB) are the most common [7]. ADHD is characterized by impairing symptoms of inattention, hyperactivity, and impulsivity [1]. Twenty percent of all individuals with ADHD are estimated to meet the diagnostic criteria for CTD, and more than sixty percent of all children and adolescents with TS are also diagnosed with ADHD [8].

Tics are believed to be related to impaired inhibition of associative and motor cortico-striato-thalamo-cortical circuits [9]. People with TD and ADHD often display hypermobility, control and regulation impairments, and difficulties in planning that share a common neurological basis, including impaired neurotransmitters activity in the basal ganglia and frontal brain regions [6,10,11,12].

Studies have indicated that comorbid ADHD places a greater burden on response inhibition in children and teens with TD than do tics, and contributes the most to the increased incidence of school problems, social difficulties, and the presence of other emotional disorders [13,14]. However, a meta-analysis showed that poor verbal and motor inhibition, inattention, and poor impulse control may be part and parcel of TD, even in the absence of any comorbid disorder [7,15]. Children and teens with TD without co-occurring ADHD may present attentional difficulties influenced by factors beyond those seen in ADHD without co-occurring TD [16]. These can include distraction from attempts to suppress the tics, and internal forms of distraction from anxiety and obsessive compulsive behaviors [16]. Consequently, tics may cause ADHD-like symptoms including behavior and attention problems that are related to the burden of the tics themselves [15].

These studies suggest that the significant impaired inhibitory performance and increased neural activity deficits in ADHD may impede patients’ ability to suppress tics and engage in therapy, thereby reducing the effect of behavioral treatments that target tics specifically.

Studies on tic suppressibility and behavioral treatment for patients with TD and co-occurring ADHD are inconclusive. Several studies have shown that individuals with comorbid TD and ADHD benefitted less from behavioral treatment than children and teens with TD without ADHD [17,18]. Sambrani et al. [7] showed that comorbid ADHD reduced successful tic suppression. However, other studies have reported that children and adolescents with TD and ADHD were able to suppress tics, and that tic suppressibility did not differ as a function of the presence or absence of ADHD comorbidity [19,20]. Yet, other studies have indicated that ADHD did not impede the effects of behavioral treatment [10,21]. Hence, further research is needed to clarify whether comorbid ADHD influences the response outcomes of behavioral treatment in children and teens. In addition, as patients become more skilled in behavioral techniques, it is conceivable that the tic suppression strategies become more automatic, and thus may lessen the attentional efforts expended to control tics. Therefore, the improvement in tics may be also associated with improvement in attention and impulse control.

OCD is characterized by the presence of obsessions, compulsions or both that cause clinically significant distress or impairment in functioning [1]. OCD was reported to be comorbid in 11% to 66% of all individuals with tics [22]. Although a tic is typically less complex than a compulsive behavior, differentiating an obsessive compulsive behavior (OCB) from a complex tic or tic-related OCD may pose a diagnostic challenge to clinicians because both disorders share premonitory stimuli preceding the reaction and repetitive behavior [1,22].

Studies by Sukhodolsky et al. [10], and McGuire et al. [17] indicated that the presence of co-occurring OCD did not moderate the response to behavioral treatment for TD, and Sambrani et al. [7] suggested that tic suppression was independent from comorbid OCD and OCB. However, Sambrani et al. [7] found that people with TD and comorbid OCD presented with a more severe type of the disorder including higher tic severity.

Although a substantial proportion of children and teens with TD have comorbid ADHD or OCD, there are only a few studies on the differential effects of behavioral treatment in this population with comorbid ADHD or OCD or studies on the effects of behavioral treatment on specific attentional and impulse control difficulties in children and teens with TD without ADHD.

Comprehensive behavioral intervention for tics (CBIT) [23] has gained robust support from randomized controlled trials (RCTs) for its efficacy in reducing tics [5,17,24] and comorbid symptoms including anxiety and obsessive compulsive symptoms [25,26]. Despite the efficacy of CBIT, the vast majority of children and teens with TD do not receive evidence-based interventions for tics [27,28], and COVID-19 has created additional obstacles to implementing psychotherapy. Computerized-based administrations can help overcome hurdles to the greater accessibility of BT [5,21] as well as CBIT in the routine care of patients at this age range with TD. A recent RCT showed that internet-based self-help CBIT (ICBIT) for children and teens supported by their parents and with minimal remote therapist intervention was also effective, with a significant improvement in comorbid symptoms six months after treatment [25]. The present study reanalyzed data from this RCT to assess the effect of internet-delivered CBIT for children and teens with TD and co-occurring ADHD or OCD compared to a similar aged group without co-occurring ADHD or OCD.

Thus, the current study aimed to assess the following: (1) whether ICBIT elicits a differential response in children and teens with comorbid ADHD compared to those with TD without ADHD; (2) the efficacy of ICBIT on specific difficulties of inattention and impulse control that are related to TD for the entire cohort; (3) whether ICBIT would elicit a differential response in children and teens with comorbid OCD compared to children and teens with TD without comorbid OCD.

## 2. Method

Data from the ICBIT trial (*n* = 38) [25] were used to assess the efficacy of ICBIT for children and teens with TD + ADHD and TD + OCD and the effect of ICBIT on ADHD-like symptoms and OCB in children and teens that were not diagnosed with full criteria of ADHD or OCD according to DSM-5 [1]. This trial was approved by the Tel-Aviv Sourasky Medical Center Research Ethics Committee and was registered in the National Institute for Health Research Portfolio Database (ClinicalTrials.gov Identifier: NCT04087616). The Consolidated Standards of Reporting Trials diagram and full inclusion and exclusion criteria were originally reported in Rachamim et al. [25]. The inclusion criteria were as follows: a primary diagnosis of TD using the DSM-5 criteria [1], a Yale Global Tic Severity Scale (YGTSS) [29] of mild to moderate, aged 7 to 18 years, and on a stable dose of psychiatric medication for at least six weeks with no planned changes (if applicable) for the upcoming six months. Patients with comorbid psychiatric disorders were included unless they required immediate intervention.

### 2.1. Assessment

An intention-to-treat principle was applied. Assessments were at baseline, post-ICBIT, 3 and 6 months post-ICBIT completion. After obtaining informed consent, eligible participants were invited to complete an online clinical assessment. Online clinical interviews were based on the Anxiety Disorders Interview Schedule, the ADIS-C/P [30]. The sample was composed of 38 children and teens (12 girls) aged 7.89–17.57 years (*M* = 11.29, *SD* = 1.97) with a principal diagnosis of TS (*n* = 29) or CTD (*n* = 9). Of these, *n* = 16 met the DSM-5 criteria for ADHD [1,30] and *n* = 11 met the DSM-5 criteria for OCD [1,30].

### 2.2. Measures

The outcomes were measured at baseline, post-intervention, and at 3 months and 6 months after randomization, and were administered by independent masked clinicians at baseline, post-intervention and at the 3- and 6-month follow-up assessments.

The primary outcomes were tic severity as measured by the YGTSS [29], a clinician-rated tic severity scale assessing the number, frequency, intensity, complexity, and interference. The following four composite scores are generated: total motor tic severity (MTS) (rated 0–25), total vocal tic severity (VTS) (0–25), total tic severity (TTS) (0–50) and impairment scale (IS) (0–50) with higher scores indicating greater severity. The YGTSS is the gold standard measure of tics and is used widely in clinical practice and research. Global assessment of symptom improvement was measured on the Clinical Global Impression—improvement scale (the CGI-I), a clinician-rated scale [31] ranging from 1 (very much improved) to 7 (very much worse). The CGI-I was used to determine improvements in TD as well as in ADHD. A score of 1 (very much improved) and 2 (much improved) was used to identify intervention responders. Functional status was assessed on the clinician-rated Global Assessment Scale for Children, the CGAS [32]. The CGAS provides a measure of global impairment and functioning over the previous month (rated 1–100).

### 2.3. Secondary Outcomes

Parents and children reported secondary outcomes online. These included The Revised Connors’ Parent Rating Scale, the CPRS-R [33] which was used to measure ADHD symptom severity. The scale produces the following 5 scores: an inattentive score (rated 0–9), a hyperactivity score (0–15), an anxiety score (0–12), a disruptive behavior score (0–39), a psychosomatic score (0–6) and a total ADHD severity score (0–144). Diagnostic status and symptom severity were assessed on the Anxiety Disorders Interview Schedule, the ADIS-C/P [30]. The ADIS child and parent versions assess anxiety, mood, and externalizing disorders in children and teens and screens for additional disorders.

OCB severity was measured through the Obsessive Compulsive Inventory; OCI-CV [34], a 21-item scale assessing symptom severity on a 3-point scale (range = 0–42).

### 2.4. Intervention

The ICBIT program consists of 9 consecutive conjoint child-caregiver modules, delivered over 9 weeks. ICBIT facilitates self-help therapy through age-appropriate texts and descriptive diagrams, animations, and video clips of clinicians demonstrating techniques. Module 1 presents psychoeducational information about tics and awareness training for tic occurrence. Module 2 covers stress management skills. Module 3 presents a function-based environmental intervention. Module 4 presents information about a competing response and how to identify the premonitory urge for the first tic. In module 5–7 the participants and their parents continue training for the second, third, fourth and the fifth tics. In module 8 the participants and their parents continue training for the sixth tic and generalization training. The patients practice various daily situations including more challenging scenarios that can exacerbate tics. Module 9 comprises skills, methods and instructions for further practice aiming at maintaining intervention gains and relapse prevention. This module was repeated once a month as a monthly booster module for the next 6 months.

### 2.5. Statistical Analysis

Baseline characteristics were compared between groups, TD + ADHD (*n* =16) or TD − ADHD (*n* = 22), TD + OCD (*n* =11) or TD − OCD (*n* = 27) with *t* tests for continuous variables and χ^2^ tests for categorical variables. Descriptive statistics were computed for demographic and clinical characteristics for the entire cohort and for each group. Differences between the groups were assessed using linear mixed effects model analyses to account for the correlations in the repeated measurements. In all linear mixed models, the comorbid condition, time, and the interaction between time and comorbid condition were added in the specification of the fixed effects. Data analyses were performed using SPSS, version 25.0 (IBM SPSS Statistics for Windows, Armonk, NY, USA). Bonferroni corrections also were applied in post hoc comparisons of marginal means. Statistical significance for all analyses was set at *p* < 0.05 for a two-tailed test.

## 3. Results

### 3.1. TD + ADHD Group vs. TD − ADHD Group

TD + ADHD group (*n* = 16) consisted of significantly more males than females compared to The TD − ADHD group (*n* = 22). The TD + ADHD group (*n* = 16) presented with more learning disabilities and social phobia disorders, and higher ADHD symptom severity as measured by the CPRS-R. The CGAS scores in the TD + ADHD group were significantly lower, suggesting that ADHD leads to greater disability than TD alone. Apart from these, no differences were found in the demographic variables (see Table 1).

The two groups were similar in the magnitude of tic reduction from baseline to post-treatment, as well as in the 3- and 6-month follow-up assessments (YGTSS TTS) *F* = 0.47, *p* = 0.70, severity of motor tics (YGTSS MTS) *F* = 0.96, *p* = 0.42, and severity of vocal tics (YGTSS VTS) *F* = 0.65, *p* = 0.58, and impairment scores (YGTSS IS) *F* = 1.62, *p* = 0.20 (see Table 2).

The improvement rate for tics (as indicated by a score of 1 and 2 on the CGI-I) was significant in both groups. At Time 2 in the TD + ADHD group 50% (8/16) improved significantly, while in the TD − ADHD group 68.18% (15/22) improved significantly. At Time 3, in the TD + ADHD group 81.25% (13/16) improved significantly while in the TD − ADHD group 72.72% (16/22) improved significantly. At Time 4, in the TD + ADHD group 87.50% (14/16) improved significantly while in the TD − ADHD group 95.45% (21/22) improved significantly, χ^2^_(4)_ = 4.31, *p* = 0.36

The two groups were similar in the magnitude of functional impairment improvement as measured by the CGAS from baseline to post-treatment, as well as in the 3- and 6-month follow-up assessments; *F* = 0.50, *p* = 0.68 (see Table 2).

According to the Parent Self-Reporting Index (CPRS-R), no significant differences were found between the groups in improving attention and concentration symptoms; *F* = 1.96, *p* = 0.13.

On the CGI-I for ADHD, those who were diagnosed with ADHD at Time 1 (*n* = 16), the improvement rate for ADHD was not significant (see Table 2).

For the whole sample (*n* = 38), there was a significant improvement on the CPRS-R total score *F* = 12.77, *p* < 0.00, as well as significant improvements on the inattentive *F* = 4.32, *p* < 0.01, hyperactivity-impulsivity *F* = 13.31, *p* < 0.00, anxiety *F* = 10.33, *p* < 0.00, and psychosomatic *F* = 5.88, *p* < 0.00, subscales from baseline to post-treatment, as well as in the 3- and 6-month follow-up assessments. However, no significant effect was found on the behavioral problems sub-scale; *F* = 3.49, *p* = 0.20.

For the whole sample (*n* = 38), there was a significant improvement on the OCI-CV total score, *F* = 11.07, *p* < 0.00.

### 3.2. TD + OCD Group vs. TD − OCD Group

No significant differences were found between the TD + OCD group (*n* = 11) and TD − OCD group (*n* = 27) for the demographic variables (see Table 1). A significant interaction between groups was found for the YGTSS TTS; *F* = 3.69, *p* = 0.02, The YGTSS TTS was significantly reduced in both groups; however, the magnitude of the total tic reduction was lower in the TD + OCD group. For the YGTSS MTS a significant interaction between group was found; *F* = 6.74, *p* = 0.00, and the magnitude of the YGTSS MTS reduction was lower in the TD + OCD group. No significant interaction was found for YGTSS VTS; *F* = 0.99, *p* = 0.40, and YGTSS IS *F* = 1.46, *p* = 0.24 (see Table 3).

The two groups were similar in the magnitude of functional impairment improvement as measured by the CGAS from baseline to post-treatment, as well as in the 3- and 6-month follow-up assessments *F* = 2.26, *p* = 0.09 (see Table 3).

The improvement rate for tic severity (as indicated by a score of 1 and 2 on the CGI-I) was significant in both groups. At Time 2, in the TD + OCD group (8/11, 72%), TD − OCD group (15/27, 55.55%), At Time 3, in the TD + OCD group (9/11, 81.81%), TD − OCD group (23/27, 85.18%), At Time 4, in the TD + OCD group (9/11, 81.81%), TD − OCD group (27/27, 100%), χ^2^_(4)_ = 13.30, *p* = 0.10 (see Table 3).

For those participants who were diagnosed with OCD at Time 1 (*n* = 11), the improvement rate on CGI-I for OCD was not significant (see Table 3).

This study consisted of a small sample size. Therefore, we were underpowered to adequately compare changes in terms of tic severity in children and teens with TD and specific comorbidities that may moderate outcomes following ICBIT, relative to children and teens without such specific comorbidities. However, to further explore treatment response in terms of comorbidity, we further classified the participants into the following 4 groups: TD + ADHD + OCD (*n* = 4), TD + ADHD − OCD (*n* = 12), TD − ADHD + OCD (*n* = 7) and TD − ADHD − OCD (*n* = 15).

The results on the CGI-I showed that at post-intervention, 13/15, 86.66% in the TD − OCD − ADHD group were rated as treatment responders. At the 3-month and 6-month follow-up assessments, 14/15, 93.33%, were rated as treatment responders. At post-intervention, 2/7, 28.57% in the TD + OCD − ADHD group were rated as treatment responders, at 3-month assessment 4/7, 57.14%, and at the 6-month assessment 6/7, 85.71% were rated as treatment responders. In the TD + ADHD − OCD group, at post-intervention, 6/12, 50.00% were rated as treatment responders, at the 3-month assessment 10/12, 83.33%, and at the 6-month assessment 11/12, 91.66% were rated as treatment responders. In the TD + OCD + ADHD group, at post-intervention, 3/4, 75.00% were rated as treatment responders. At 3-month and 6 months follow-up assessments, 4/4, 100%, were rated as treatment responders. Figure 1 presents the Mean of baseline (time 1), post-intervention (time 2), 3-month (time 3) and 6-month (time 4) scores on the Yale Global Tic Severity Scale Total Tic Score measure of TD + OCD + ADHD, TD + ADHD − OCD, TD + OCD − ADHD, TD − ADHD − OCD.

## 4. Discussion

In this post hoc reanalysis of our previous RCT [25], we aimed to explore the effects of ICBIT on children and teens with TD with and without ADHD or OCD and assessed changes in ADHD and OCD symptomatology over a 3- and 6-month follow-up period. The results showed that ICBIT had a comparable effect on tic reduction in participants with ADHD. These data are consistent with studies indicating that children and teens with ADHD can benefit comparably from CBIT as was found in previous studies [10,21]. However, the current study showed that although both tics and functional impairment of the participants with co-occurring ADHD improved, they had lower CGAS scores, indicating greater impairment, compared to the participants without ADHD. In addition, the TD + ADHD group had significantly more learning disabilities, social phobia and higher ADHD symptom severity based on the CPRS-R. These results suggest that comorbid ADHD was associated with greater disability than TD alone. Our findings are consistent with previous studies indicating that co-existing ADHD often causes more psychosocial impairment for children and adolescents with TD than the severity of the tics themselves [8,13,16].

The current data showed improvement on the inattention, psychosomatic, anxiety, and hyperactivity-impulsivity scores based on parental reports (CRPS-R). These findings suggest that the tic reduction may lessen the burden placed by tics on attentional resources [15] and thereby further contribute to the improvement on the CRPS-R scores. Alternatively, the reduction in inattention, psychosomatic, anxiety, and hyperactivity-impulsivity symptoms may be attributed to a possible effect of CBIT on brain plasticity, specifically in the cortico-striato-thalamo-cortical circuits [9,35]. These two studies showed that CBIT may promote neurocognitive changes and normalization of deficits in cortico-striato-thalamo-cortical circuits in individuals with TD, which may account for our findings of an improvement in inattention, psychosomatic, anxiety and impulsivity-hyperactivity symptoms. As ADHD and TD share both common and different brain regions [11,13,36], it is conceivable that the improvements on the CRPS-R scores may be related to the positive effects of the core therapeutic techniques of CBIT; namely, tic awareness training, stress management and tic suppression by implementing habit reversal skills. These techniques may have a positive effect on certain neurological circuits associated with the impaired inhibitory performance and with attentional and sensory regulation mechanisms underlying both ADHD and TD [9,13].

High rates of comorbid obsessive compulsive symptoms were reported in children and teens with TD [7]. Furthermore, there is evidence that both OCD and TD share common underlying neurological circuits [36]. The ICBIT program yielded significant improvement in obsessive and compulsive symptoms. These findings may suggest that several components of ICBIT, may enhance regulatory capabilities associated with both the improvement of OCB and TD. However, the current study suggests that the participants with TD and comorbid OCD gained less from ICBIT than the participants without comorbid OCD on the total tic severity and the motor tic severity YGTSS scales. These results are in line with Sambrani et al. [7], suggesting that OCD may further complicate interventions for tics. On the other hand, other studies [10,17] showed that the presence of comorbid OCD did not moderate treatment response. These inconclusive findings suggest that further studies are needed to examine the effect of comorbid OCD in order to ensure better treatment outcomes.

Overall, the results of the current study indicate that ICBIT may be effective for children and teens with TD and ADHD. However, the results show that an OCD comorbid condition may attenuate treatment outcomes. 

Despite the fact that the majority of children and teens with TD experience clinically significant reduction in TD symptoms following behavioral treatment, the effectiveness of the treatment is not sufficient to relieve all the tic symptoms [5]. As was previously reported [25], the results on the CGI-I for the total sample showed that at post ICBIT 81.57% were rated as treatment responders with further improvement during the follow-up period (91.66%, at 3-month follow-up and 94.44%, at 6-month follow-up assessment). However, this study suggests that the percent classified as responders for children with TD with comorbid ADHD or/and OCD, at post intervention, was lower relative to teens and children without OCD or/and ADHD. In addition, at post intervention, the percent of children and teens classified as responders for children with TD + OCD + ADHD was the lowest. There is therefore a need to understand the comorbidities that might attenuate treatment response in order to identify ways to augment or improve behavioral treatment for pediatric TD. Whilst the current findings provide support for ICBIT in treating TD with OCD and ADHD comorbidity, the results also imply that individuals with OCD comorbid condition may gain less from ICBIT. These findings highlight the need to determine if outcomes might be improved for children and teens with TD and OCD comorbid condition by tailoring interventions to address comorbid psychiatric disorders.

The use of the internet as a therapeutic platform can provide a safe and effective alternative solution to dealing with accessibility problems. The outbreak of the COVID-19 pandemic has posed many new challenges to traditional face-to-face therapy. ICBIT may become a standard method for the dissemination of evidence-based therapy for children and teens with TD and comorbid ADHD and OCD.

However, the results should be considered in light of the limitations of this study, which consisted of a small sample. Further, since children and teens with ADHD or OCD who required immediate treatment were not included, it is likely that severe ADHD or OCD may further complicate interventions targeting tics [6]. The post hoc examination of TD without comorbid OCD or ADHD, TD and OCD with or without ADHD, and TD and ADHD with or without OCD within this sample resulted in relatively small samples for each group. Analyses are likely to be affected by limited power. Further research with larger clinical sub-samples across each comorbid group is necessary to confirm outcomes reported in this trial with greater statistical power. Future research investigating the impact of comorbidity on treatment response with large samples, including children with complex comorbid conditions is warranted to optimize responses to ICBIT in pediatric TD. Overall, these positive results are promising, and suggest that ICBIT is compatible for children and teens with comorbid ADHD or OCD. However, it is important to study strategies that may enhance therapeutic outcomes to optimize the treatment of pediatric TD and comorbid disorders that may moderate response to therapy.

## 5. Clinical Implications

Assessing the effect of comorbid disorders on ICBIT is the first step toward enhancing the ability of children and teens with TD to adhere to required treatment assignments that are likely to positively affect treatment gains. The current findings suggest that tics, OCD and ADHD symptoms should be carefully assessed and be the target of behavioral interventions in young people with TD.

Therapy for TD with comorbid OCD, OCB or ADHD should be tailored to individual difficulties and needs, with various modalities that focus on tics and comorbid symptoms. These interventions can include adaptations to young people with ADHD, such as immediate rewards, and allowing for more frequent breaks to encourage the child’s motivation and regulation [19,25,26,37]. The current findings, alongside previous studies, are indicative of the positive effects of tic-focused behavioral treatment on comorbidity, and may suggest that ICBIT can be offered as a first step in the treatment of children and teens with tics and mild comorbid symptoms, as it has a positive effect on concomitant symptomatology in addition to tic severity reduction. The next steps, if required, may include more specific cognitive and behavioral interventions, such as exposure and response prevention for OCD and OCB, cognitive techniques for improving anxiety symptoms, and specific behavioral modifications for comorbid ADHD [37].

## Figures and Tables

**Figure 1 jcm-11-00045-f001:**
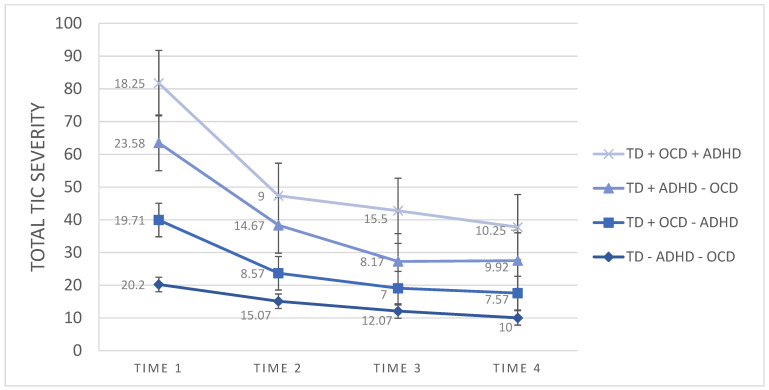
Mean of baseline (time 1), post-intervention (time 2), 3-month (time 3) and 6-month (time 4) scores on Yale Global Tic Severity Scale Total Tic Score measure of TD + OCD + ADHD, TD + ADHD − OCD, TD + OCD − ADHD, TD − ADHD − OCD.

**Table 1 jcm-11-00045-t001:** Baseline Demographic and Clinical Characteristics.

Measure		TD + OCD (*n* = 11)	TD − OCD (*n* = 27)	Statistic	*p*	TD + ADHD (*n* = 16)	TD − ADHD (*n* = 22)	Statistic	*p*
Age, mean (SD)		12.00 (2.37)	10.97 (1.72)	*t*_(36)_ = −1.61	0.11	11.84 (2.47)	10.90 (1.43)	*t*_(36)_ = −1.48	0.14
Gender, *n* (%)	Males	7 (63.63%)	19 (70.37%)	χ^2^_(1)_ = 0.16	0.68	15 (93.75%)	11 (50.00%)	χ^2^_(1)_ = 8.26	0.00
	Females	4 (36.36%)	8 (29.62%)			1 (6.25%)	11 (50.00%)		
Current medication use, *n* (%)	No	7 (63.63%)	22 (81.48%)	χ^2^_(6)_ = 8.53	0.20	9 (56.25%)	20 (90.90%)	χ^2^_(6)_ = 8.43	0.20
Past psycho-therapy experience, *n* (%)	No	6 (54.54%)	7 (25.92%)	χ^2^_(5)_ = 4.22	0.51	3 (18.75%)	10 (45.45%)	χ^2^_(5)_ = 7.54	0.18
Tic disorder, *n* (%)	CTD	3 (27.27%)	6 (22.22%)	*t*_(7)_ = 0.00	0.38	3 (18.75%)	6 (27.27%)	*t*_(7)_ = 0.79	0.45
	TS	8 (72.72%)	21 (77.77%)	*t*_(27)_ = 1.25	0.58	13 (81.25%)	16 (72.72%)	*t*_(27)_ = −1.21	0.23
SAD, *n* (%)		3 (27.27%)	3 (11.11%)	χ^2^_(1)_ = 2.54	0.11	2 (12.5%)	4 (18.18%)	χ^2^_(1)_ = 0.11	0.73
GAD, *n* (%)		5 (45.45%)	10 (37.03%)	χ^2^_(1)_ = 2.78	0.42	7 (43.75%)	8 (36.36%)	χ^2^_(1)_ = 1.25	0.53
SPD, *n* (%)		2 (18.18%)	7 (25.92%)	χ^2^_(1)_ = 0.25	0.61	7 (43.75%)	2 (9.09%)	χ^2^_(1)_ = 6.15	0.01
SP, *n* (%)		1 (9.09%)	3 (11.11%)	χ^2^_(1)_ = 0.07	0.78	-	4 (18.18%)	χ^2^_(1)_ = 3.17	0.07
OCD, *n* (%)		4 (36.36%)	7 (25.92%)	χ^2^_(1)_ = 2.66	0.44	4 (25.00%)	7 (31.81%)	χ^2^_(1)_ = 2.66	0.44
Dysthymia, *n* (%)		1 (9.09%)	2 (7.40%)	χ^2^_(1)_ = 0.37	0.53	1 (6.25%)	2 (9.09%)	χ^2^_(1)_ = 0.06	0.80
Enuresis, *n* (%)		1 (9.09%)	1 (3.70%)	χ^2^_(1)_ = 1.04	0.30	1 (6.25%)	1 (4.54%)	χ^2^_(1)_ = 0.10	0.74
Encopresis, *n* (%)		-	1 (3.70%)	χ^2^_(1)_ = 0.41	0.51	1 (6.25%)	-	χ^2^_(1)_ = 1.41	0.23
LD, *n* (%)		2 (18.18%)	4 (14.81%)	χ^2^_(1)_ = 0.07	0.79	5 (31.25%)	1 (4.54%)	χ^2^_(1)_ = 4.96	0.02
SMD, *n* (%)		1 (9.09%)	5 (18.51%)	χ^2^_(1)_ = 0.52	0.47	4 (25.00%)	2 (9.09%)	χ^2^_(1)_ = 1.76	0.18
YGTSSMTS, mean (SD)		19.28 (4.85)	21.63 (6.22)	*t*_(36)_ = 1.16	0.25	15.63 (2.36)	15.18 (3.66)	*t*_(36)_ = −0.42	0.67
YGTSSVTS, mean (SD)		13.73 (2.61)	16.04 (3.14)	*t*_(36)_ = 2.14	0.38	6.06 (4.38)	4.77 (5.26)	*t*_(36)_ = −0.79	0.43
YGTSS TTS, mean (SD)		5.45 (5.76)	2.26 (4.61)	*t*_(36)_ = −0.11	0.91	22.25 (5.67)	19.95 (6.00)	*t*_(36)_ = −1.19	0.24
YGTSS IS, mean (SD)		31.82 (14.70)	28.89 (15.27)	*t*_(36)_ = −0.54	0.59	31.88 (16.41)	28.18 (14.01)	*t*_(36)_ = −0.74	0.46
OCI-CV (parent), mean (SD)		4.82 (4.57)	4.48 (4.24)	*t*_(36)_ = −0.21	0.83	-	-	-	-
OCI-CV, mean (SD)		13.73 (5.40)	10.74 (6.54)	*t*_(36)_ = −1.33	0.19	-	-	-	-
CPRS-R, mean (SD)						41.75 (16.05)	21.73 (14.42)	*t*_(36)_ = −4.02	0.00
	Inattentive, mean (SD)					6.19 (2.58)	2.45 (2.52)	*t*_(36)_ = −4.45	0.00
	Hyperactivity, mean (SD)					9.50 (3.68)	5.95 (3.98)	*t*_(36)_ = −2.79	0.00
	Anxiety, mean (SD)					6.94 (2.74)	3.64 (2.96)	*t*_(36)_ = −3.49	0.00
	Disruptive behavior, mean (SD)					15.31 (11.14)	6.55 (6.02)	*t*_(36)_ = −3.12	0.00
	Psychosomatic, mean (SD)					3.81 (2.90)	3.14 (2.90)	*t*_(36)_ = −0.73	0.46
CGAS, mean (SD)		66.64 (10.49)	69.15 (12.78)	*t*_(36)_ = 0.57	0.56	62.00 (10.12)	73.09 (11.38)	*t*_(36)_ = 3.10	0.00

ADHD attention deficit hyperactivity disorder, SAD separation anxiety disorder, SPD social phobia disorder, GAD general anxiety disorder; SP specific phobia, OCD obsessive compulsive disorder and subclinical OCD, LD learning disabilities, SMD sensory modulation disorder. MTS motor tic score, VTS vocal tic score, YGTSS TTS Yale Global Tic Severity Scale Total Tic Score, YGTSS IS Yale Global Tic Severity Scale Impairment Score, CPRS-R Conner’s Rating Scales-Revised, CGAS Children’s Global Assessment Scale, OCI-CV obsessive compulsive inventory.

**Table 2 jcm-11-00045-t002:** Baseline (time 1), post-intervention (time 2) and 3-month (time 3) and 6-month (time 4) scores on measures of TD and ADHD.

	TD + ADHD (*n* = 16)				TD − ADHD (*n* = 22)					
	Time 1 Mean (SD)	Time 2 Mean (SD)	Time 3 Mean (SD)	Time 4 Mean (SD)	Time 1 Mean (SD)	Time 2 Mean (SD)	Time 3 Mean (SD)	Time 4 Mean (SD)	Time Effect (*F* Value, *p*)	Interaction (*F* Value, *p*)
YGTSS MTS	15.61 (0.97)	9.55 (1.17)	7.42 (1.10)	7.86 (0.99)	15.95 (0.86)	10.71 (1.02)	7.83 (0.98)	6.47 (0.88)	*F* = 81.27, *p* = 0.00	*F* = 0.96, *p* = 0.42
YGTSS VTS	7.55 (1.30)	4.51 (1.00)	3.04 (0.91)	2.36 (0.68)	4.78 (1.15)	3.05 (0.87)	2.25 (0.80)	1.93 (0.60)	*F* = 7.94, *p* = 0.00	*F* = 0.65, *p* = 0.58
YGTSS TTS	23.66 (1.61)	15.66 (2.02)	11.50 (1.95)	11.36 (1.75)	20.82 (1.43)	14.08 (1.79)	11.23 (1.69)	9.86 (1.75)	*F* = 56.02, *p* = 0.00	*F* = 0.47, *p* = 0.70
YGTSS IS	35.00 (3.27)	12.10 (2.36)	6.43 (2.35)	2.73 (1.70)	27.82 (3.27)	12.10 (2.36)	3.89 (2.09)	4.95 (1.51)	*F* = 40.61, *p* = 0.00	*F* = 1.62, *p* = 0.20
CPRS-R	44.27 (3.86)	35.38 (3.93)	31.61 (3.77)	29.25 (3.65)	23.08 (3.41)	17.52 (3.48)	18.82 (3.33)	15.29 (3.65)	*F* = 14.96, *p* = 0.00	*F* = 1.96, *p* = 0.13
Inattentive	6.44 (0.63)	5.66 (0.70)	5.61 (0.72)	5.00 (0.62)	2.47 (0.56)	1.91 (0.62)	2.30 (0.64)	1.41 (0.54)	*F* = 4.27, *p* = 0.01	*F* = 0.20 *p* = 0.89
Hyperactivity	10.00 (0.90)	7.83 (0.83)	7.33 (0.89)	6.10 (0.87)	6.04 (0.80)	4.47 (0.73)	4.39 (0.79)	3.95 (0.76)	*F* = 14.51, *p* = 0.00	*F* = 1.09, *p* = 0.36
Anxiety	7.16 (0.58)	5.16 (0.63)	4.38 (0.66)	3.26 (0.53)	3.60 (0.58)	2.52 (0.56)	2.82 (0.58)	1.87 (0.53)	*F* = 12.40, *p* = 0.00	*F* = 2.31, *p* = 0.09
Disruptive Behavior	16.33 (2.30)	13.94 (2.27)	11.50 (2.07)	12.14 (2.15)	7.91 (2.04)	6.47 (2.01)	6.95 (1.83)	5.98 (1.89)	*F* = 5.34, *p* = 0.00	*F* = 1.64, *p* = 0.19
Psychosomatic	4.33 (0.71)	2.77 (0.67)	2.77 (0.74)	1.84 (0.54)	3.04 (0.63)	2.13 (0.59)	2.34 (0.65)	1.84 (0.54)	*F* = 6.03, *p* = 0.00	*F* = 0.42, *p* = 0.73
CGAS	61.27 (2.54)	67.23 (2.73)	71.49 (2.78)	74.13 (2.94)	72.51 (2.24)	79.73 (2.37)	82.82 (2.44)	84.00 (2.56)	*F* = 18.08, *p* = 0.00	*F* = 0.50, *p* = 0.68
CGI-I TD	-	1.93 (0.19)	1.56 (0.21)	1.37 (0.15)	-	1.86 (0.16)	1.55 (0.19)	1.45 (0.13)	*F* = 91.87, *p* = 0.00	*F* = 0.28, *p* = 0.83
CGI-I ADHD	-	-	-	-	-	3.50 (0.19)	3.18 (0.20)	3.12 (0.22)	*F* = 1.69, *p* = 0.19	-

MTS motor tic score, VTS vocal tic score, YGTSS TTS Yale Global Tic Severity Scale Total Tic Score, YGTSS IS Yale Global Tic Severity Scale Impairment Score, CPRS-R Conner’s Rating Scales-Revised, CGAS Children’s Global Assessment Scale, CGI-I Clinical Global Impression rating scale.

**Table 3 jcm-11-00045-t003:** Baseline (time 1), post-intervention (time 2) and 3-month (time 3) and 6-month (time 4) scores on measures of TD and OCD.

	TD + OCD (*n* = 11)				TD − OCD (*n* = 27)					
	Time 1 Mean (SD)	Time 2 Mean (SD)	Time 3 Mean (SD)	Time 4 Mean (SD)	Time 1 Mean (SD)	Time 2 Mean (SD)	Time 3 Mean (SD)	Time 4 Mean (SD)	Time Effect (*F* Value, *p*)	Interaction (*F* Value, *p*)
YGTSS MTS	15.46 (1.15)	7.15 (0.75)	8.03 (1.36)	7.09 (1.25)	15.96 (0.78)	11.35 (0.75)	7.62 (0.90)	7.17 (0.83)	*F* = 69.78, *p* = 0.00	*F* = 6.74, *p* = 0.00
YGTSS VTS	6.38 (1.58)	3.24 (1.21)	3.16 (1.09)	2.42 (0.81)	5.82 (1.07)	3.86 (0.80)	2.33 (0.71)	1.98 (0.54)	*F* = 6.74, *p* = 0.00	*F* = 0.99, *p* = 0.40
YGTSS TTS	21.84 (1.94)	13.00 (2.37)	13.56 (2.55)	10.43 (1.45)	22.17 (1.32)	15.60 (1.61)	10.97 (1.70)	10.43 (1.45)	*F* = 44.02, *p* = 0.00	*F* = 3.69, *p* = 0.02
YGTSS IS	33.07 (4.45)	11.53 (3.26)	6.43 (2.85)	3.070 (2.09)	30.00 (3.03)	14.25 (2.15)	4.33 (1.87)	3.98 (1.37)	*F* = 35.31, *p* = 0.00	*F* = 1.46, *p* = 0.24
CGAS	64.92 (2.28)	75.05 (3.62)	77.54 (3.66)	81.14 (3.72)	68.82 (2.28)	73.98 (2.45)	77.98 (2.46)	79.06 (3.72)	*F* = 22.29, *P* = 0.00	*F* = 2.26, *p* = 0.09
OCI-CV	14.07 (1.67)	9.15 (1.70)	9.23 (1.84)	5.47 (1.67)	10.71 (1.13)	6.67 (1.16)	6.89 (1.25)	4.86 (1.11)	*F* = 10.80, *P* = 0.00	*F* = 0.35, *p* = 0.78
CGI-I TD	-	1.36 (0.20)	1.72 (0.25)	1.36 (0.18)	-	2.11 (0.13)	1.48 (0.17)	1.44 (0.12)	*F* = 78.99, *p* = 0.00	*F* = 2.50, *p* = 0.06
CGI-I OCD	-	-	-	-	-	1.80 (0.24)	1.80 (2.00)	1.60 (0.26)	*F* = 1.79, *p* = 0.18	-

MTS motor tic score, VTS vocal tic score, YGTSS TTS Yale Global Tic Severity Scale Total Tic Score, YGTSS IS Yale Global Tic Severity Scale Impairment Score, OCI-CV obsessive compulsive inventory, CGAS Children’s Global Assessment Scale, CGI-I Clinical Global Impression rating scale.

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
