# Peer review of "Acute and Long-Term Effects of an Internet-Based, Self-Help Comprehensive Behavioral Intervention for Children and Teens with Tic Disorders with Comorbid Attention Deficit Hyperactivity Disorder, or Obsessive Compulsive Disorder: A Reanalysis of Data from a Randomized Controlled Trial"

_jcm, 2021, doi:10.3390/jcm11010045_

Round 1
Reviewer 1 Report
This study is a re-analysis of the Rachamim et al trial published last year (ref 25 in the reference list). The rationale for the study is well presented and the ms is generally well written. The sample size is very small and probably inadequate to address the research question. The statistical analysis is quite basic and could be improved. In my opinion, this paper does not constitute a significant advance but it does provide some useful data. Some specific suggestions for improvement:
Title, abstract and introduction. These sections should clearly indicate that this study is a re-analysis of a previously published trial. They should also indicate that the analyses are post-hoc. For example, the last paragraph of the introduction should read: “In this study, we re-analyzed data from our previous trial (25) and aimed to (1) assess whether….” (or similar text).
Line 126 (when study 25 is first mentioned), please add the sample size of the study.
Lines 129-131. The authors write: However, to the best of our knowledge no study has compared the efficacy of ICBIT… Because this is a post-hoc analysis which is probably underpowered, I do not think that the authors can firmly conclude anything about the efficacy of the treatment for these patient sub-groups. Please soften the language to ‘explore’ or similar.
Statistical analyses. The last assessment carried forward method is a poor choice. There is consensus amongst statisticians that this should be avoided. Instead, consider using linear mixed models in STATA or similar software, which can cope with missing data under the assumption that these are missing at random.
Results, line 221. This reads as if the TD – ADHD group had higher CPRS-R scores, which is obviously not the case. Please reword.
Discussion. Please start this section with something like: “In this post-hoc reanalysis of our previous RCT (25), we aimed to explore …..” (or similar text).
The authors found that individuals with OCD comorbidity improved less on the primary outcome measure (YGTSS) and write that these findings are in line with those of another study (ref 7). However, they should also list much better powered studies that did not find this. In other words, I do not think that the results of the current analyses can be taken as a definitive answer to the question.
Additionally, the paragraph starting in line 324 contradicts the previous paragraph (“the results indicate that ICBIT may be effective for children with TS with ADHD or OCD disorders”). In the previous paragraph we read that OCD may associated with worse outcomes.
Author Response
Dear Editor,
Thank you for your thorough and constructive review of our manuscript as well as providing us with the opportunity to re-submit our manuscript with revisions. We would like to convey our thanks to the reviewers for their detailed and insightful remarks helping us improve our manuscript. We have revised our manuscript according to the remarks of the reviewers. We believe that consequently, the paper is more informative and coherent as a result of these revisions.
Title, abstract and introduction. These sections should clearly indicate that this study is a re-analysis of a previously published trial. They should also indicate that the analyses are post-hoc. For example, the last paragraph of the introduction should read: “In this study, we re-analyzed data from our previous trial (25) and aimed to (1) assess whether….” (or similar text).
Thank you for your thorough and constructive review of our manuscript as well as providing us with the opportunity to re-submit our manuscript with revisions.
We strongly agree with this comment and revised the abstract. We previously mentioned in the method section that the data was collected from the CBIT trial, and we added this comment also in the discussion section.
Line 126 (when study 25 is first mentioned), please add the sample size of the study.
Added.
Lines 129-131. The authors write: However, to the best of our knowledge no study has compared the efficacy of ICBIT… Because this is a post-hoc analysis which is probably underpowered, I do not think that the authors can firmly conclude anything about the efficacy of the treatment for these patient sub-groups. Please soften the language to ‘explore’ or similar.
Revised.
Statistical analyses. The last assessment carried forward method is a poor choice. There is consensus amongst statisticians that this should be avoided. Instead, consider using linear mixed models in STATA or similar software, which can cope with missing data under the assumption that these are missing at random.
We strongly agree with this comment and after consulting a statistician, we ran the data as was suggested and changed the tables and the results section accordingly. The results and trends obtained using linear mixed models were comparable to our previous results.
Results, line 221. This reads as if the TD – ADHD group had higher CPRS-R scores, which is obviously not the case. Please reword.
Thank you. Revised.
Discussion. Please start this section with something like: “In this post-hoc reanalysis of our previous RCT (25), we aimed to explore …..” (or similar text).
Revised.
The authors found that individuals with OCD comorbidity improved less on the primary outcome measure (YGTSS) and write that these findings are in line with those of another study (ref 7). However, they should also list much better powered studies that did not find this. In other words, I do not think that the results of the current analyses can be taken as a definitive answer to the question.
We strongly agree with this comment. We elaborated on this and discussed the study of Sukhodolsky et al., and McGuire et al in this paragraph.
Additionally, the paragraph starting in line 324 contradicts the previous paragraph (“the results indicate that ICBIT may be effective for children with TS with ADHD or OCD disorders”). In the previous paragraph we read that OCD may associate with worse outcomes.
Thank you for your accurate review, corrected.
Reviewer 2 Report
This study reported on the effects of an internet-delivered, self-help Comprehensive Behavioral Intervention for Tics (ICBIT) in children and adolescents with Tic Disorders (TD) with and without comorbid diagnoses of ADHD or OCD.
Since ADHD or OCD are frequently comorbid with TD, it is clinically important to investigate the effects of ICBIT in consideration of them. Therefore, this study is very meaningful.
However, I think the following points need to be considered.
Although the effects of the presence or absence of ADHD or OCD were investigated in this study, 38 children and adolescents with TD can be classified into 4 groups such as TD + ADHD + OCD, TD + ADHD-OCD, TD-ADHD + OCD and TD-ADHD-OCD (TD only). Since the number of each group is small, it seems difficult to compare the effects among the 4 groups. However, I would like to know at least the number of TD + ADHD + OCD and/or the number of TD only to further examine the effects. And if the authors can discuss a little about TD + ADHD + OCD and TD only, please add it.
TD is stated to have been diagnosed using DSM-5 criteria. According to the notation of DSM-5, TS corresponds to Tourette's Disorder, and CTD corresponds to Persistent (Chronic) Motor or Vocal Tic Disorder. Were comorbidities also diagnosed by DSM-5? For example, can learning disabilities and social phobia be considered the same as specific learning disorders and social anxiety disorders in DSM-5?
Is it possible to divide Table 1 into two tables such as Characteristic in Baseline with and without ADHD and Characteristics in Baseline with and without OCD? If Table 1 cannot be divided into two due to the limited number of tables, it would be appropriate to place the ADHD which was mentioned earlier in the text, on the left side of Table 1. Also, Gender is shown using n (%) instead of mean (SD).
Tables 1 and 3 showed the evaluation of obsessive-compulsive symptoms by OCI-CV (Obsessive-Compulsive Inventory), but it was not mentioned at all in the text. If OCI-CV is shown in the tables, please describe it as one of the measurement tools, and indicate the results and meaning of the evaluation. According to Table 3, many children and adolescents with TD had obsessive-compulsive symptoms even if they were not diagnosed with comorbid OCD, and obsessive-compulsive symptoms are alleviated with the implementation of ICBIT regardless of the presence or absence of OCD. Please discuss the effects of ICBIT from this point also.
In Discussion, the authors stated that "the results of the current study indicate that ICBIT may be effective for children and teens with TD with ADHD or OCD disorders." However, can the effects of comorbid ADHD or OCD be really considered the same?
Although it is a small point, "higher CGAS" on P9 L296 seems to be a mistake of "lower CGAS".
Author Response
Dear Editor,
Thank you for your thorough and constructive review of our manuscript as well as providing us with the opportunity to re-submit our manuscript with revisions. We would like to convey our thanks to the reviewers for their detailed and insightful remarks helping us improve our manuscript. We have revised our manuscript according to the remarks of the reviewers. We believe that consequently, the paper is more informative and coherent as a result of these revisions.
Thank you for your thorough and constructive review of our manuscript as well as providing us with the opportunity to re-submit our manuscript with revisions.
Although the effects of the presence or absence of ADHD or OCD were investigated in this study, 38 children and adolescents with TD can be classified into 4 groups such as TD + ADHD + OCD, TD + ADHD-OCD, TD-ADHD + OCD and TD-ADHD-OCD (TD only). Since the number of each group is small, it seems difficult to compare the effects among the 4 groups. However, I would like to know at least the number of TD + ADHD + OCD and/or the number of TD only to further examine the effects. And if the authors can discuss a little about TD + ADHD + OCD and TD only, please add it.
As you suggested, the sample is small so we did not compare the effects on each group. For example, in the group TD+ADHD+OCD there were only 4 children. Due to the small sample, we used descriptive statistics (see table 3 and figure1) to address these interesting data.
TD is stated to have been diagnosed using DSM-5 criteria. According to the notation of DSM-5, TS corresponds to Tourette's Disorder, and CTD corresponds to Persistent (Chronic) Motor or Vocal Tic Disorder. Were comorbidities also diagnosed by DSM-5? For example, can learning disabilities and social phobia be considered the same as specific learning disorders and social anxiety disorders in DSM-5?
Yes, comorbidities were also diagnosed by DSM-5 criteria. Please see p. 7, line 5 (Method section) and line 6 (Assessment section).
Is it possible to divide Table 1 into two tables such as Characteristic in Baseline with and without ADHD and Characteristics in Baseline with and without OCD? If Table 1 cannot be divided into two due to the limited number of tables, it would be appropriate to place the ADHD which was mentioned earlier in the text, on the left side of Table 1.
Due to the limited number of figures and tables we placed the ADHD data on the left side of table 1 as was suggested.
Also, Gender is shown using n (%) instead of mean (SD).
Thank you for your accurate review, corrected.
Tables 1 and 3 showed the evaluation of obsessive-compulsive symptoms by OCI-CV (Obsessive-Compulsive Inventory), but it was not mentioned at all in the text. If OCI-CV is shown in the tables, please describe it as one of the measurement tools, and indicate the results and meaning of the evaluation. According to Table 3, many children and adolescents with TD had obsessive-compulsive symptoms even if they were not diagnosed with comorbid OCD, and obsessive-compulsive symptoms are alleviated with the implementation of ICBIT regardless of the presence or absence of OCD. Please discuss the effects of ICBIT from this point also.
Added as requested in the measurement tools' section as well as in the discussion.
In Discussion, the authors stated that "the results of the current study indicate that ICBIT may be effective for children and teens with TD with ADHD or OCD disorders." However, can the effects of comorbid ADHD or OCD be really considered the same?
Revised.
Although it is a small point, "higher CGAS" on P9 L296 seems to be a mistake of "lower CGAS".
Thank you for your accurate review, corrected.
Round 2
Reviewer 1 Report
The authors have been responsive and done their best to improve the ms. The study is still severely limited by its sample size and for this reason its impact on the field will be modest.
Author Response
Thank you for your constructive review of our manuscript. We went through the paper once again to improve style and some minor check spell, and hope that the paper is now ready for publication.